# Effect of cosmetic hair treatment and natural hair colour on hair testosterone concentrations

**Julia K. Preinbergs**[1]*, **Inger Sundström-Poromaa**[2], **Elvar Theodorsson**[1], **Jakob O. Ström** [1,3], **Edvin Ingberg**[4]

1 Division of Clinical Chemistry and Pharmacology, Department of Biomedical and Clinical Sciences, Faculty of Medicine and Health Sciences, Linköping University, Linköping, Sweden, 2 Department of Women's and Children's Health, Faculty of Medicine, Uppsala University, Uppsala, Sweden, 3 Department of Neurology, Faculty of Medicine and Health, Örebro University, Örebro, Sweden, 4 Department of Infectious Diseases, Faculty of Medicine and Health, Örebro University, Örebro, Sweden

* julia.preinbergs@regionostergotland.se

## Abstract

### Purpose

Testosterone analysis in hair allows for retrospective evaluation of endogenous testosterone concentrations, but studies devoted to investigating confounders in hair testosterone analysis have hitherto been scarce. The current study examined the stability of testosterone concentrations between two hair samples collected three months apart and investigated two potential confounding factors: natural hair colour and cosmetic hair treatments.

### Methods

Testosterone was analysed with an in-house radioimmunoassay with a limit of detection adequate for the purpose.

### Results

The testosterone concentrations from the two samplings, at baseline and three months later, had an intra-individual correlation of moderate strength (rho = 0.378, p<0.001, n = 146). Hair treatment, such as colouring or bleaching, seemed to increase testosterone concentrations (p = 0.051, n = 191, and in a paired analysis in a subset of the cohort p = 0.005, n = 24), while no effect of natural colour in untreated hair (p = 0.133) could be detected.

### Conclusion

The current results suggest that cosmetic hair treatments need to be considered in hair testosterone analyses and demonstrate the utility of a radioimmunoassay to reliably measure testosterone concentrations in small hair samples in women.

**Data Availability Statement:** All relevant data are within the paper and its Supporting information files.

**Funding:** This work was financially supported by the Swedish Research Council project K2013-99X-22269-01-3, the County Council of Östergötland, and the Family Planning Foundation. The funders had no role in the study design, data collection and analysis, decision to publish, or preparation of the manuscript.

**Competing interests:** "JOS received advisory board fees from Bayer Pharmaceuticals during 2016 and 2019, but the advisory boards were not in any way related to the subject of the current paper. JKP, ISP, ET and EI have no relevant financial or non-financial interests to disclose. This does not alter our adherence to PLOS ONE policies on sharing data and materials."

## Introduction

Current methods for testosterone analysis in the clinical setting, mainly blood or saliva samples, are fraught with difficulties in determining the free hormone concentration and need multiple samplings to incorporate the dynamics in hormone concentrations. Measurement of steroid hormones in hair extracts has been proposed as a useful method for clinical research to describe long-term retrospective endogenous steroid hormone concentrations, with a time-frame of one month per centimetre of hair growth [1]. Common practice in hair analysis is to cut and analyse the most proximal three centimetres of hair collected at the posterior vertex of the scalp. The mechanisms of hormone incorporation into the hair are still being elucidated, but the currently favoured theory is that the free, biologically active fraction of total hormone concentrations in the plasma is incorporated in actively growing hair through passive diffusion from capillaries surrounding the hair follicle [2, 3]. Other possible mechanisms of hormone incorporation from sebum and sweat, as well as local hormone production, are not fully understood.

Cortisol is currently the predominant steroid hormone analysed in hair, along with other hormones regulated by the HPA-axis, followed by androgens. A significant, albeit moderate, correlation between hair cortisol concentrations and salivary cortisol concentrations has been described, with similar findings for hair testosterone [4–6]. The intra-individual stability of cortisol in repeated hair measurements, described by Stadler *et al.*, has been argued to support the usefulness of hair hormone analyses [7]. Possible confounders, such as hair treatment, sex, and medication use, have been defined for cortisol in hair, while such investigations regarding testosterone have been lacking [8, 9].

In this study we examined the stability in hair testosterone concentrations in repeated samples cut three months apart and investigated two potential confounding factors in a cohort of women from a randomised controlled trial of an oral contraceptive [10]. The effect of cosmetic hair treatment (such as colouring or bleaching) on testosterone concentrations and the association between natural hair colour and testosterone concentrations were explored.

## Materials and methods

### Subjects

Participants were 202 women included in a multi-centre, randomised, placebo-controlled and double-blind trial studying different aspects of an oral contraceptive containing 1.5 mg of estradiol and 2.5 mg of nomegestrol acetate [10]. The participants were randomized in a 1:1 ratio to either treatment with the oral contraceptive or placebo. Hair was sampled twice, at baseline before randomization (at the posterior vertex of the scalp) and at follow-up three months later (adjacent to the first sampling area). The participants were 18 to 41 years old, physically healthy, and non-obese (BMI < 30 kg/m2). In accordance with clinical routine for treatment with combined oral contraceptives, participants with prior history of venous thromboembolism, systolic blood pressure > 140 mmHg or diastolic blood pressure > 90 mmHg, known dyslipidaemia, migraine with focal symptoms, inflammatory bowel disease, first-degree relatives with cardiovascular disease at a young age, previous cancer and previous pancreatitis were excluded. All participants gave written informed consent prior to inclusion. The study was approved by the regional Ethical Review Board, EPN 2013/161 Uppsala, Sweden. Information on current and previous (within six months) hair treatment was recorded at each hair sampling. The natural hair colour was assessed by looking at the intact hair samples of participants that reported no hair treatment during the relevant study period and categorised according to shade (light, medium, dark).

## Hormonal analysis

Hair segments of three centimetres were analysed. Each hair sample was pulverised and extracted with methanol. Testosterone was measured using an in-house competitive radioimmunoassay in vacuum-concentrated extracts as previously described in detail [11]. Sample weights were on average 9.7 mg (SD 2.1). All hair samples were analysed in duplicate and in the same assay. Seventy individuals donated hair samples that were large enough to be divided into triplicate samples prior to pulverisation (to reduce the variability from extraction and measurement); these samples were extracted and analysed in parallel, and the average testosterone concentration was calculated after completion of the analysis.

## Statistical analysis

Level of significance for all statistical analyses was set at $\alpha = 0.05$, and tests were two-sided. The distribution of hair testosterone results was positively skewed. The correlation between the two hair samplings (at baseline and at follow-up after three months) was analysed using Spearman's correlation coefficient. In the original study by Lundin *et al.* no effect of the combined oral contraceptive on hair testosterone concentrations was detected why no regard was taken to whether the participants had been randomized to placebo or to hormonal contraceptive in the main correlation analysis in the current study. A secondary correlation analysis with only participants from the placebo group was also performed. The effect of hair treatment, such as hair colouring or bleaching, on testosterone concentrations was explored using the Mann-Whitney U-test with hair samples from the baseline sampling (n = 191). Further, some of the participants (n = 24) changed from untreated to treated hair, or the opposite, between the two hair samplings. Although it was not planned before the study, related-samples Wilcoxon signed rank test was used to explore the intra-individual effect of hair treatment compared to natural untreated hair. To check for age-related confounding, whether participants at a certain age were more predisposed to colouring or bleaching their hair, the age distribution within the categories (treated and untreated) was compared with the Mann-Whitney U-test. The natural hair colour was divided into three categories (light, medium and dark shade). Natural darker red hair was included in the medium shade category and natural lighter red hair was included in the light shade category. There were no samples with predominantly grey hair. Hair testosterone concentrations in the natural hair colour categories was analysed using the non-parametric Jonckheere-Terpstra test, exploring whether there was a trend of increasing or decreasing hormone concentrations in relation to the shade of natural hair colour. All analyses were conducted using the IBM SPSS statistics 28.0 package.

# Results

## Hormone stability between samplings

At baseline hair samples were cut from 191 women, and at the final visit a repeated hair sample was cut in 154 women, of which eight participants did not contribute with a hair sample at baseline. The intra-individual stability in testosterone between inclusion and three months later was explored and revealed a moderately sized significant correlation between hair samplings (whole cohort: rho = 0.378, p<0.001, n = 146; only placebo group: rho = 0.331, p = 0.003, n = 77), see Fig 1. Twenty-four individuals in the cohort changed hair treatment between inclusion and three months later, meaning that some individuals with natural hair at inclusion had coloured or bleached their hair during the study, thus the hair sample at the follow-up was treated. Other individuals switched from treated hair at baseline to outgrown and

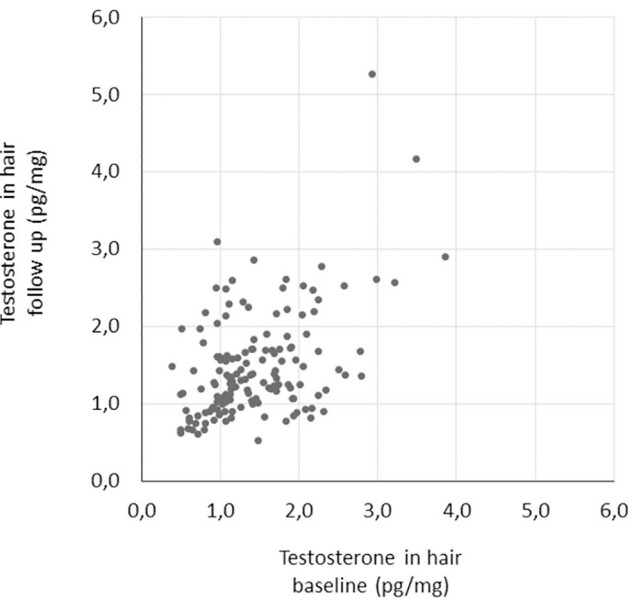

**Fig 1. Testosterone in hair cut at inclusion (baseline) and a second sample cut three months later (follow-up) from the 146 participants that donated hair on both occasions, correlated with moderate strength (Spearman's rho = 0.378, p<0.001).**

untreated hair at follow-up. A repeated correlation analysis with these 24 individuals excluded rendered a slightly stronger association (rho = 0.425, p<0.001, n = 122).

## Effect of hair treatments on testosterone concentrations

Almost half of the study participants (43%) had treated hair at baseline. Mann-Whitney U-test for the distribution across categories (natural hair compared to treated hair) showed that treated hair samples had higher testosterone concentrations, with the test statistic verging on significance (p = 0.051; Fig 2).

As previously described, twenty-four individuals changed hair treatment between inclusion and three months later. The mean within-individual difference between inclusion and three months later in these twenty-four persons was 0.50 pg/mg (median 0.33, range -1.45–3.58 pg/mg) with the higher concentrations found in treated hair (p = 0.005, paired samples Wilcoxon test), see Fig 3.

Median age in the cohort was 24 years, mean age was 24.2 years. No statistically significant difference in participant age between the hair treatment categories could be found (p = 0.598, n = 191).

## Effect of natural hair colour on hormone concentrations

Testosterone concentrations in untreated hair samples from the baseline sampling were not significantly associated with the individuals' natural hair colour (p = 0.133, n = 108), see Fig 4. Testosterone concentrations in the categories were: light hair median 1.06 pg/mg (mean 1.17 pg/mg, n = 28), medium hair median 1.26 pg/mg (mean 1.52 pg/mg, n = 48), dark hair median 1.25 pg/mg (mean 1.37 pg/mg, n = 32).

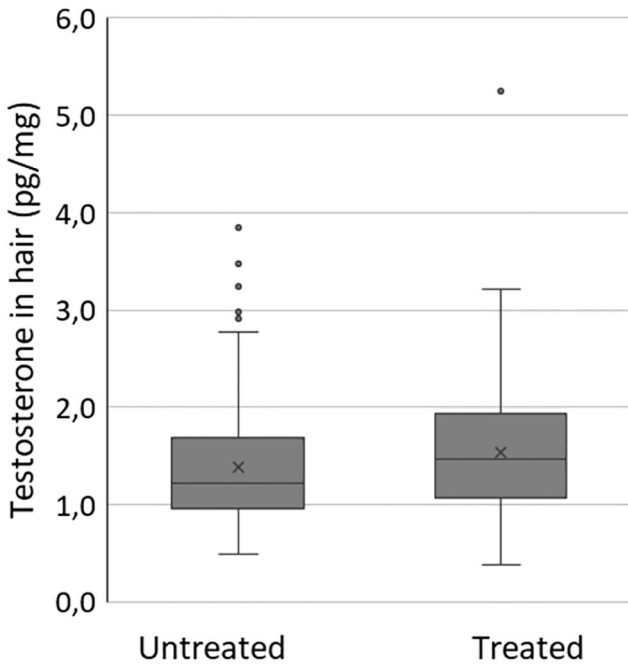

**Fig 2. Comparison of hair testosterone concentrations in natural untreated (n = 108) and treated (coloured or bleached, n = 83) hair samples from the baseline sampling.** Median in untreated hair was 1.21 pg/mg and in treated hair 1.47 pg/mg, Mann-Whitney U test between the two groups was bordering significance (p = 0.051).

## Discussion

In this observational study, testosterone concentrations in hair samples cut three months apart correlated with moderate strength. Cosmetic hair treatment had a moderate but significant effect on testosterone hair concentrations, while no effect of natural colour of untreated hair could be detected. The intra-individual correlation of testosterone in hair over a three-month period is in line with previous research and indicates that hair hormones reflect endogenous hormone exposure over time [4].

There is a paucity of studies exploring relations between sociodemographic, physiological, and analytical factors and endogenous testosterone levels in hair. Only a small number of studies describe statistical analyses where some factors are controlled for, but to our knowledge, no studies with a systematic approach to potential confounders in hair testosterone analysis have been published [12–14]. Van Manen *et al.* mentions that no effect of artificial hair colouring or hair washing frequency on hair testosterone levels was detected, but no data is shown [15]. Cortisol in hair has on the other hand gained considerable interest, and factors that affect hair cortisol levels include age, sex, BMI, certain medication and illnesses, hair washing frequency, hair treatments with chemicals or heat, exposure to sunlight or ultraviolet light, and perhaps also smoking, alcohol use, and socioeconomic status [8, 9, 16–19]. In a large cohort (n = 3675, aged 59–83 years) lower cortisol concentrations were found in treated hair in a mutually adjusted linear regression analysis, which contrasts the results for testosterone in the current study [18]. Also, in an experimental setting when aliquots of hair sampled on one occasion from one single person were dyed rendered lower cortisol levels compared to native untreated hair [20].

A possible explanation for increased hormone concentrations in treated hair in the current study is that chemical processing of the hair such as colouring or bleaching could interfere

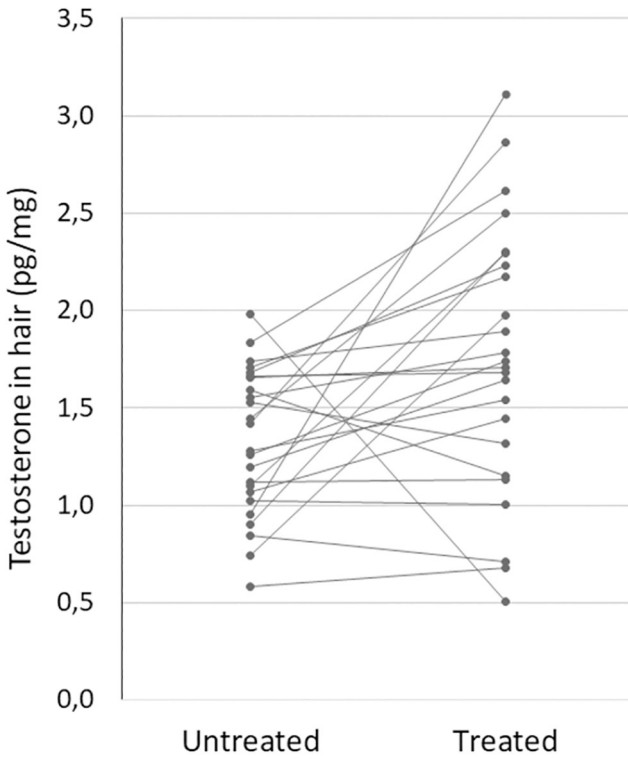

**Fig 3. Twenty-four individuals changed hair treatment between the two hair samplings.** The hair testosterone concentrations were grouped according to the treatment category instead of sampling time (baseline or follow-up). On average the testosterone concentrations were 0.5 pg/mg higher in the treated hair category (p = 0.005, paired samples Wilcoxon test).

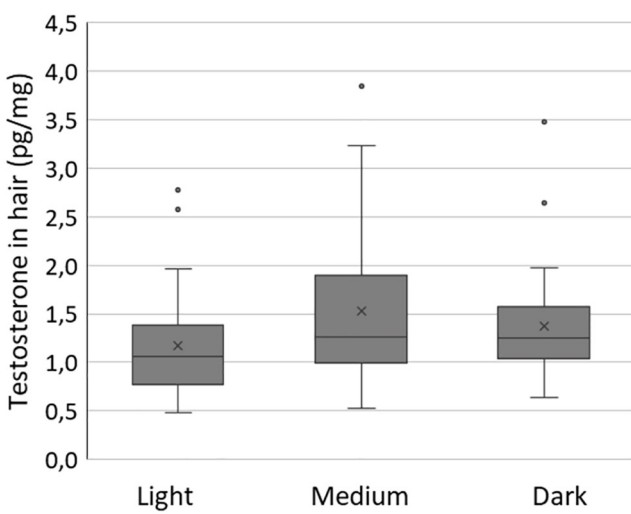

**Fig 4. Testosterone concentrations in natural untreated baseline hair samples grouped according to natural hair colour (n = 108).** Although lightly pigmented hair had nominally lower median testosterone concentrations, the effect of natural hair colour did not reach statistical significance.

with the binding of the antibody to testosterone in hair extracts and to the radioligand. In our in-house radioimmunoassay the relationship between hormone concentrations and the measured radioactivity is inverse, which would result in falsely elevated concentrations. Whether this effect is specific for the testosterone antibody we are using (described in detail by Slezak *et al.* [11]), or a generic phenomenon remains to be established. Another plausible explanation is that the process of colouring or bleaching inflicts changes in the hair matrix allowing for a larger quantity of the hormones to be extracted from the pulverised hair. Interestingly, Nagata *et al.* described significantly higher serum testosterone concentrations in women that have been colouring their hair for some years compared to those who never coloured their hair, but the effect was not preserved after controlling for covariates [21]. Further on, although not relevant for the current study, some of the artificial colour from coloured hair was occasionally present in the hair extracts, visible to the naked eye. When using colorimetric immunosorbent methods, this extracted colour could possibly confound the measurement of absorbance. An advantage of using gamma radiation as marker in our study is that it is not affected by the colour of the extracts.

Nominally lower testosterone concentrations in lightly pigmented hair compared to the two darker shades were found in the current study, but no significant association across categories of natural hair colour could be identified. The potential effect of natural hair colour on steroid hormone content in hair remains ambiguous. Higher cortisol concentrations in individuals with natural black hair compared to individuals with lighter shades has been reported [8, 16, 22], as well as no differences in hair cortisol concentrations in relation to natural hair colour [23]. The potential effect of melanin on hormone content in hair was examined by Voegel *et al.* where pigmented hair was compared to grey hair within the same individuals (n = 18) with no significant intra-individual difference between pigmented and grey hairs [24].

A strength of the current study was that the participants were young (aging between 18 to 41 years), non-obese and of generally good health, which decreased the risk of confounding factors such as abnormal BMI or chronic medication. However, limitations in the current study may be the lack of information on ethnical background, hair washing frequency, and use of hair styling products before hair sampling. Further, we made no distinctions between hair bleaching, hair toning, or hair colouring. Some participants had both bleached and coloured their hair within the study period, thus, a distinction between hair treatments was not deemed feasible with regard to the size of the cohort and insufficient information regarding the details of repeated hair treatments (e.g. first bleached, later coloured, or bleached and coloured at the same time). It is possible that hair colour (natural and/or artificially coloured) would interfere less in LC-MS/MS measurements than in immunoassays. The radioimmunoassay used in this study is not as specific as other published mass-spectrometric methods such as LC-MS/MS. However, the current study would not have been able to be performed with current published LC-MS/MS methods because of the substantial proportions of non-detected testosterone concentrations in samples from women [12, 24]. The testosterone-like immunoreactivity detected by our assay has been shown to provide physiologically relevant measurement results and the low detection limit makes it possible to reliably measure testosterone in smaller samples of hair [11]. Asking for larger hair samples from women may result in reluctance in participating in clinical studies due to cosmetic concerns. All participants in both the placebo and the treatment-arm were included in the current correlation analysis because the original study by Lundin *et al.* did not detect any changes in hair testosterone in the treatment arm [10], which is in line with previous research [12].

With hair hormone analyses becoming increasingly used, possible interferences in the measurement methods need to be evaluated. Our results show significant effects of hair treatment on testosterone concentrations in hair. We also show that testosterone concentrations display

some degree of intra-individual stability in hair over a three-month period, which is in line with findings in hair cortisol research. This supports the use of hormone analyses in hair as a measure of endogenous hormone exposure over time.

## Supporting information

**S1 Checklist. STROBE statement—Checklist of items that should be included in reports of observational studies.**
(DOCX)

**S1 Data.**
(XLSX)

## Acknowledgments

Expert advice from Dr Agota Malmborg and Dr Cecilia Lundin is gratefully acknowledged.

## Author Contributions

**Conceptualization:** Julia K. Preinbergs, Inger Sundström-Poromaa.

**Formal analysis:** Julia K. Preinbergs, Jakob O. Ström, Edvin Ingberg.

**Funding acquisition:** Inger Sundström-Poromaa, Elvar Theodorsson.

**Investigation:** Julia K. Preinbergs, Inger Sundström-Poromaa.

**Resources:** Elvar Theodorsson.

**Supervision:** Elvar Theodorsson, Jakob O. Ström, Edvin Ingberg.

**Writing – original draft:** Julia K. Preinbergs.

**Writing – review & editing:** Inger Sundström-Poromaa, Elvar Theodorsson, Jakob O. Ström, Edvin Ingberg.

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
