## [Decision Letter · Decision Letter 0]

23 Jun 2023

PONE-D-23-10423Effect of cosmetic hair treatment and natural hair colour on hair testosterone concentrationsPLOS ONE

Dear Dr. Ström,

Thank you for submitting your manuscript to PLOS ONE. After careful consideration, we feel that it has merit but does not fully meet PLOS ONE’s publication criteria as it currently stands. Therefore, we invite you to submit a revised version of the manuscript that addresses the points raised during the review process.

We look forward to receiving your revised manuscript.

Kind regards,

Samuel Asamoah Sakyi, Ph.D

Academic Editor

PLOS ONE

Journal Requirements:

"This work was financially supported by the Swedish Research Council project K2013-99X-22269-01-3, the County Council of Östergötland, and the Family Planning Foundation. "

"JOS received advisory board fees from Bayer Pharmaceuticals during 2016 and 2019, but the advisory boards were not in any way related to the subject of the current paper. JKP, ISP, ET and EI have no relevant financial or non-financial interests to disclose."

Reviewers' comments:

Reviewer's Responses to Questions

**Comments to the Author**

1. Is the manuscript technically sound, and do the data support the conclusions?

Reviewer #1: Partly

Reviewer #2: Yes

2. Has the statistical analysis been performed appropriately and rigorously? 

Reviewer #1: Yes

Reviewer #2: Yes

3. Have the authors made all data underlying the findings in their manuscript fully available?

Reviewer #1: No

Reviewer #2: Yes

4. Is the manuscript presented in an intelligible fashion and written in standard English?

Reviewer #1: Yes

Reviewer #2: Yes

5. Review Comments to the Author

Reviewer #1: The study entitled “Effect of cosmetic hair treatment and natural hair colour on hair testosterone concentrations” submitted for publication in Plos One took the hair samples from the initial clinical study and reevaluated the hair testosterone concentrations solely. Thereby hair testosterone concentrations were compared between two different sample collection time points (3 months apart) and analyzed for potential differences resulting from natural hair color or cosmetic hair treatment. The results are described sufficiently, and the statistical tests were appropriate. However, overall, the study can only be considered preliminary as only observations were provided that do not allow (yet) for a final conclusion. I miss some experiments that would allow for detailed conclusions, e.g., applying an additional analytical method (e.g., mass spectrometry) that would allow differentiating between actual in vivo effects or artificial analytical increases of the testosterone concentrations. Also, additional in vitro experiments would have allowed the differentiation of different hair treatment procedures against untreated hair. At least regarding the use of mass spectrometry methods, the authors discuss in detail why such analysis where not feasible. But without further experiments, the data and results seem too small to be published in a full-length original research article. Publication of the presented observations might be better suited as a short communication or technical note maybe in a different journal.

Some minor comments:

Introduction: Please go into more detail about why testosterone should be measured in hair.

Fig 2 and 3 could be combined in just one figure.

Reviewer #2: The manuscript is well written and the analysis adequate. A drawback might be the used methodology (immunoassay) compared to studies that have used LC-MS/MS. But this fact is extensively discussed in the discussion. Therefore I recommend the manuscript for publication.

6. PLOS authors have the option to publish the peer review history of their article (what does this mean?). If published, this will include your full peer review and any attached files.

Reviewer #1: No

Reviewer #2: No

---

## [Author Response · Author response to Decision Letter 0]

1 Aug 2023

Response to reviewers

Reviewer #1: The study entitled “Effect of cosmetic hair treatment and natural hair colour on hair testosterone concentrations” submitted for publication in Plos One took the hair samples from the initial clinical study and reevaluated the hair testosterone concentrations solely. Thereby hair testosterone concentrations were compared between two different sample collection time points (3 months apart) and analyzed for potential differences resulting from natural hair color or cosmetic hair treatment. The results are described sufficiently, and the statistical tests were appropriate. However, overall, the study can only be considered preliminary as only observations were provided that do not allow (yet) for a final conclusion. I miss some experiments that would allow for detailed conclusions, e.g., applying an additional analytical method (e.g., mass spectrometry) that would allow differentiating between actual in vivo effects or artificial analytical increases of the testosterone concentrations. Also, additional in vitro experiments would have allowed the differentiation of different hair treatment procedures against untreated hair. At least regarding the use of mass spectrometry methods, the authors discuss in detail why such analysis where not feasible. But without further experiments, the data and results seem too small to be published in a full-length original research article. Publication of the presented observations might be better suited as a short communication or technical note maybe in a different journal.

Reply: We appreciate that Reviewer #1 correctly describes the context of our observational study with repeated hair samples from 146 women. A future study with an experimental study design is needed to assess and provide more data on the effects of in vivo and peri-analytical factors on hair testosterone concentrations in women. 

Some minor comments:

Introduction: Please go into more detail about why testosterone should be measured in hair.

Fig 2 and 3 could be combined in just one figure.

Reply: The first paragraph of the introduction has been altered according to the comment. Figure 3 consists of a paired subset of the participants (n=24), while Figure 2 consists of the whole cohort (n=191), which is the reason why the figures cannot be combined.

Reviewer #2: The manuscript is well written and the analysis adequate. A drawback might be the used methodology (immunoassay) compared to studies that have used LC-MS/MS. But this fact is extensively discussed in the discussion. Therefore I recommend the manuscript for publication.

Reply: Thank You for appreciating our work.

---

## [Decision Letter · Decision Letter 1]

7 Sep 2023

Effect of cosmetic hair treatment and natural hair colour on hair testosterone concentrations

PONE-D-23-10423R1

Dear Dr. Strom,

We’re pleased to inform you that your manuscript has been judged scientifically suitable for publication and will be formally accepted for publication once it meets all outstanding technical requirements.

Kind regards,

Samuel Asamoah Sakyi, Ph.D

Academic Editor

PLOS ONE

Additional Editor Comments (optional):

Reviewers' comments:

Reviewer's Responses to Questions

**Comments to the Author**

1. If the authors have adequately addressed your comments raised in a previous round of review and you feel that this manuscript is now acceptable for publication, you may indicate that here to bypass the “Comments to the Author” section, enter your conflict of interest statement in the “Confidential to Editor” section, and submit your "Accept" recommendation.

Reviewer #1: (No Response)

Reviewer #2: All comments have been addressed

2. Is the manuscript technically sound, and do the data support the conclusions?

Reviewer #1: Partly

Reviewer #2: Yes

3. Has the statistical analysis been performed appropriately and rigorously? 

Reviewer #1: Yes

Reviewer #2: Yes

4. Have the authors made all data underlying the findings in their manuscript fully available?

Reviewer #1: Yes

Reviewer #2: Yes

5. Is the manuscript presented in an intelligible fashion and written in standard English?

Reviewer #1: Yes

Reviewer #2: Yes

6. Review Comments to the Author

Reviewer #1: I understand that the authors would like to publish their results in the current form and would like to avoid further experiments. Still, in my personal opinion, the content of the manuscript remains too preliminary for publication as a full-length article.

Reviewer #2: (No Response)

7. PLOS authors have the option to publish the peer review history of their article (what does this mean?). If published, this will include your full peer review and any attached files.

Reviewer #1: No

Reviewer #2: No

---

## [Editor Report · Acceptance letter]

15 Sep 2023

PONE-D-23-10423R1 

Effect of cosmetic hair treatment and natural hair colour on hair testosterone concentrations 

Dear Dr. Ström:

I'm pleased to inform you that your manuscript has been deemed suitable for publication in PLOS ONE. Congratulations! Your manuscript is now with our production department. 

Kind regards, 

on behalf of

Dr. Samuel Asamoah Sakyi 

Academic Editor

PLOS ONE